# Antiphospholipid Syndrome in Pregnancy: New and Old Pathogenetic Mechanisms

**DOI:** 10.3390/ijms24043195

**Published:** 2023-02-06

**Authors:** Silvia D’Ippolito, Greta Barbaro, Carmela Paciullo, Chiara Tersigni, Giovanni Scambia, Nicoletta Di Simone

**Affiliations:** 1Dipartimento di Scienze della Salute della Donna, del Bambino e di Sanità Pubblica, Fondazione Policlinico Universitario Agostino Gemelli, Istituto di Ricovero e Cura a Carattere Scientifico (IRCCS), L. go A. Gemelli 8, 00168 Rome, Italy; 2Dipartimento di Scienze della Vita e Sanità Pubblica, Università Cattolica del Sacro Cuore, L. go A. Gemelli 8, 00168 Rome, Italy; 3Department of Biomedical Sciences, Humanitas University, Via Rita Levi Montalcini 4, 20072 Milan, Italy; 4IRCCS Humanitas Research Hospital, Via Manzoni 56, 20089 Milan, Italy

**Keywords:** antiphospholipid syndrome, obstetrical APS, pathogenesis, thrombosis, inflammation

## Abstract

The antiphospholipid syndrome (APS) is a systemic autoimmune disorder characterized, according to the Sydney criteria, by the persistent presence of autoantibodies directed against phospholipid-binding proteins associated with thrombosis and/or obstetrical complications. The most frequent complications in obstetric antiphospholipid syndrome are recurrent pregnancy losses and premature birth due to placental insufficiency or severe preeclampsia. In recent years, vascular APS (VAPS) and obstetric APS (OAPS) have been described as two different clinical entities. In VAPS, antiphospholipid antibodies (aPL) interfere with the mechanisms of coagulation cascade and the ‘two hit hypothesis’ has been suggested to explain why aPL positivity does not always lead to thrombosis. OAPS seems to involve additional mechanisms, such as the direct action of anti-β2 glycoprotein-I on trophoblast cells that can lead to a direct placental functional damage. Furthermore, new actors seem to play a role in the pathogenesis of OAPS, including extracellular vesicles, micro-RNAs and the release of neutrophil extracellular traps. The aim of this review is to investigate the state-of-the-art antiphospholipid syndrome pathogenesis in pregnancy, in order to provide a comprehensive overview of both old and new pathogenetic mechanisms involved in this complex disease.

## 1. Introduction

The antiphospholipid syndrome (APS) is a systemic autoimmune disorder whose diagnostic criteria were revised in 2006 at the Sydney workshop in Australia. Concisely, APS is characterized by persistent positivity for antiphospholipid antibodies [aPL, including lupus anticoagulant (LA), anticardiolipin (aCL) and anti-β2-glycoprotein1 antibodies (anti-β 2GPI)], thrombotic events and/or severe pregnancy morbidity [1,2,3,4]. Beyond the APS-related thrombotic and pregnancy complications, additional clinical conditions, such as cutaneous (livedo reticularis, cutaneous ulcers) haematological (thrombocytopenia, hemolytic anemia), cardiac (cardiac valvular disease), nephrological and neurological manifestations are associated to APS [5].

The reported annual incidence and prevalence of APS in adults of 2.1 per 100,000 and 50 per 100,000, respectively, with both incidence and prevalence similar in both sexes [6].

APS can be defined as “primary APS” when it is isolated, or “secondary APS” when it occurs associated to other autoimmune diseases, mainly systemic lupus erythematous (SLE) [7]. Cervera et al. found that the 53.1% of patients had “primary APS”, 36.2% had systemic lupus erythematosus, 5.0% had lupus-like syndrome, 2.2% had primary Sjögren’s syndrome, 1.8% had rheumatoid arthritis, 0.7% had systemic sclerosis, 0.7% had systemic vasculitis and 0.5% had dermatomyositis [8]. Noteworthy, several non-autoimmune conditions, such as infections, malignancies and medications can determine a positivity for aPL [9,10]. Patients with primary APS and patients with secondary APS associated to SLE, have a higher prevalence and a higher level of IgA aCL and IgA anti-β 2GPI, when compared to patients with SLE or other autoimmune diseases and without APS. Moreover, these antibodies demonstrate an important correlation with thrombotic events [11]. aPLs positivity can worsen the prognosis of several diseases, which was well demonstrated by Ruaro and coll., who described a correlation between antiphospholipid antibody positivity and pulmonary embolism events in sarcoidosis patients [12]. In the recent years, a further classification of APS distinguishes vascular APS (VAPS) and obstetric APS (OAPS) [13]. VAPS is mainly characterized by venous, arterial and small vessel thrombotic events in different organs. OAPS is characterized by serious pregnancy complications including recurrent pregnancy losses, ≥1 fetal loss beyond 10 weeks of gestation, preeclampsia and placental insufficiency. All of these complications in pregnancy are not necessarily associated to placental thrombotic phenomena [14].

Most patients can show both clinical manifestations related to OAPS and VAPS, however some patients can show just vascular or obstetric symptoms [15]. In the present review, we will focus on OAPS, and in particular on the pathogenesis of pregnancy complications observed in the syndrome. The aim of this review is to investigate the state-of-the-art antiphospholipid syndrome pathogenesis in pregnancy with an innovative approach examining all of the known pathogenetic mechanisms contributing to the obstetric clinical manifestations of this complex disease, hoping that this work could provide the reader with an overview as complete as possible to better understand the clinical and therapeutic implications.

## 2. Classification Criteria of Antiphospholipid Syndrome

The initial APS diagnostic criteria, inherited by the 8th International APS Symposium of Sapporo (Japan) were revised in 2006, at the Sydney workshop in Australia. The revised diagnostic criteria include the presence of at least one of the clinical and one of the laboratory criteria (Table 1). The Sydney classification included the addition of anti-β2 glycoprotein-I antibodies (anti-β 2GPI) among the laboratory criteria, which were not considered in the previous classification, and a subclassification of APS in four different subgroups, according to different aPL positivity combinations [16]. Of note, there have been reported cases of APS showing a low titer of aPL or only one positive test, and therefore not in agreement with the Sydney criteria. These cases are classified as non-criteria OAPS (NC-OAPS) and four different subsets, based to the degree of fulfilling the standard diagnostic criteria, have been recently proposed [17].

There is also a severe form of APS termed catastrophic APS (CAPS) characterized by widespread thrombotic microangiopathy and multiorgan failure. It affects about 1% of APS patients and has a fulminant course with >40% mortality [18].

## 3. Obstetric Clinical Manifestations

The pregnancy morbidity in OAPS encloses >3 consecutive and spontaneous early miscarriages before 10 weeks of gestation, at least one unexplained fetal death after the 10th week of gestation of a morphologically normal fetus, a premature birth before the 34th week of gestation of a normal neonate due to eclampsia or severe pre-eclampsia or placental insufficiency [19].

Early pregnancy loss has been found as the most frequent obstetrical complication in APS [20]. In 2019, the European Register of Obstetrical APS (EUROAPS) published data on a cohort of 1000 obstetrical APS patients from 30 European hospitals reporting a prevalence of 38.6% for miscarriage (38.6%), 18.1% for early preeclampsia and 16.1% for early intrauterine fetal growth restriction (IUGR) [21]. APS contributes to about 15% of recurrent pregnancy loss (RPL) cases [22]. Furthermore, Saccone et al. assessed the obstetrical complications rate in pregnant women with primary APS. They divided women into two groups, according to the positivity of the laboratory tests: group I with more than one positivity to antibodies and group II with only one positivity to antibodies. In particular, triple-positive women in group I showed a lower live birth rate and a significant incidence rate of IUGR. No differences were found in the rate of preeclampsia, venous thromboembolism, pregnancy loss, preterm birth, stillbirth and neonatal mortality, between the two groups [23].

A recent metanalysis by Walter et al. tried to identify possible predictors of pregnancy complications and the magnitude of their effect in women with APS. They found that previous thrombosis was associated with a decreased risk of live birth (OR 0.60, *p* < 0.01, I^2^ = 40%), and an increased risk of neonatal mortality (OR 15.19, *p* < 0.01, I^2^ = 0%), antenatal or postpartum thrombosis (OR 6.26, *p* < 0.01, I^2^ = 0%). This group also showed an increased risk of delivering a small for gestational age neonate (SGA) (OR 2.60, *p* = 0.01, I^2^ = 0%). In addition, patients with a double or triple aPL positivity had a decreased risk of live birth (OR 0.66, *p* < 0.01, I^2^ = 0%), an increased risk of SGA (OR 1.86, *p* = 0.01, I^2^ = 43%), of preterm birth (OR 1.35, *p* < 0.01, I^2^ = 49%) and preeclampsia (OR 2.43, *p* = 0.02, I^2^ = 35%). Finally, patients with lupus anticoagulant positivity showed an increased risk of preeclampsia (OR 2.10, *p* = 0.02, I^2^ = 48%), SGA (OR 1.78, *p* < 0.01, I^2^ = 0%) and preterm birth (OR 3.56, *p* = 0.01, I^2^ = 48%). Therefore, (i) previous thrombosis, (ii) laboratory double or triple positivity and (iii) lupus anticoagulant positivity can be considered the most important predictors of adverse pregnancy outcomes [24].

In a case-control study, Marchetti et al. determined the incidence of aPLs, 6 months after delivery in women with a history of non-severe preeclampsia (PE) or severe PE. They concluded that severe PE was a distinct entity from non-severe PE and was mainly correlated with the plasma titers of anti-β2GP1 [25].

Furthermore, the presence of antiphosphatidylserine/prothrombin antibodies (aPS/PT antibodies) in APS patients seems to be associated to pregnancy complications. These antibodies target other cell membrane antigens that are not routinely required to define APS. APS pregnant women with aPS/PT antibodies have a higher rate of pregnancy complications, such as intrauterine fetal death, preterm labor, preeclampsia, and IUGR, as compared to APS pregnant women without aPS/PT antibodies. Moreover, the titer of aPS/PT IgG antibodies is inversely proportional to the neonatal weight at birth and multiple markers of vascular injury are described in IUGR placentas with aPS/PT positive patients [26]. Of interest, it has been reported that aPS/PT positivity represents a risk factor of RPL even in the absence of a clear diagnosis of APS, hence suggesting an independent pathogenic activity of these antibodies [27].

## 4. Pathogenesis of the Prothrombotic State in APS

The main mechanisms involved in the aPL-mediated thrombosis include inhibition of the anticoagulant cascade, reduction of the fibrinolytic activity, procoagulant activation of endothelial cells, increased platelet aggregation and complement activation [28]. However, in spite of the persistent aPL positivity, patients only occasionally show thrombotic events. This evidence led to theorize the ‘second hit hypothesis’, according to which, despite the presence of an induced procoagulant phenotype which may be genetically determined and finally defines the so-called first hit, an additional inciting factor, namely, “second hit,” is necessary for the clinical manifestations of APS [29]. Currently, infections are considered one of the most relevant environmental factors responsible for both aPL production and APS development. Furthermore, non infective conditions, such as obesity, smoking, immobility, estrogen pharmacological stimulation, other genetic prothrombotic diseases have been recognized as “second hit” [30].

Antiphospholipid antibodies can lead to the thrombotic event by affecting several mechanisms. They can inhibit the anticoagulant cascade, enhance hypofibrinolysis, interact and activate both endothelial cells and platelets and induce the complement cascade (Figure 1). It has been shown that antiphospholipid antibodies inhibit the protein C system [31,32] and interfere with the protein S and protein Z activation [33,34,35]. Anti-β2GPI also interfere with the binding of β2GPI to factor XI, hence decreasing the generation of factor X and an increase the thrombin production [36,37]. Furthermore, they impair fibrinolytic activity, possibly inducing high levels of inhibitor of plasminogen type 1 (PAI-1) or reducing the release of the activator of tissue plasminogen (tPA) by endothelial cells (ECs). The imbalance between PAI-1/tPA has been suggested as a crucial event in the APS-related thrombosis [38]. It is known that lipoprotein a (Lpa) impairs the fibrinolytic activity inhibiting tPA and increasing the expression of PAI-1 in ECs [39]. Of interest, Atsumi et al. reported increased levels of plasmatic Lpa in APS patients [40,41]. This might explain the reduction of fibrinolysis in these patients. Furthermore, the existence of antibodies specifically interacting with tPA could represent a further mechanism of reduced fibrinolysis [42].

Antiphospholipid antibodies can directly interact with ECs and monocytes through specific receptors. It has been reported that aPLs induce tissue factor (TF) and endothelin 1 expression in ECs and monocytes [43]. The interaction between aPLs and target cells are related to an externalization of phosphatidylserine, which is essential for the binding [28]. Annexin A2 is expressed on ECs and the monocyte acting as receptor for tPA and plasminogen. Once adhered, the complex β2GPI/anti-β2GPI on ECs interacts with Toll-like receptors (TLR)-4 and cause the signaling cascade activation. TLR-4 acts also as an adaptor for annexin 2 [44]. In addition, megalin/gp330 is an endocytic receptor that internalizes multiple ligands, including apolipoprotein E and B100. It has been reported that megalin is a receptor of the β2GPI molecule and the β2GPI-phospholipid complex [45]. Recently, apolipoprotein E receptor 2 (ApoER2) has been proposed as a receptor for β2GPI on platelets. β2GPI/anti-β2GPI antibody complex blocks ApoER2 leading to a loss of adhesion of platelets to collagen [46]. As further mechanisms, β2GPI can bind GPIbα, a subunit of the platelet adhesion molecule. This enables anti-β2GPI antibodies binding hence determining the activation of platelets, resulting both in thromboxane production, activation of the phosphoinositol-3 kinase (PI3K)/Akt pathway, both mechanisms leading to platelet adhesion and aggregation. These pathways lead to the translocation of nuclear factor kB (NF-kB) that brings it to the transcription of several genes, finally causing an amplification of the production of TF and adhesion molecules [47].

Animal models have suggested the complement activation in the pathophysiology of thrombosis in APS, with more recent data from human studies confirming the interaction between the coagulation and complement pathways. Activation of the complement cascade via aPL can cause cellular injury and promote coagulation via multiple mechanisms [48].

## 5. Pathogenesis of the Placental Damage in APS

Thrombotic events were initially considered the major pathological cause of pregnancy complications in OAPS [49]. It is now accepted that thrombosis alone cannot explain the pregnancy morbidity observed in OAPS. Indeed, the observation that not all OAPS placentas showed thrombotic phenomena, lead to suggest further direct pathogenic actions of aPL on the placental tissue [18]. Accordingly, animal models have demonstrated the direct aPL-mediated functional damage on placental tissue and/or the inflammatory processes involving the complement cascade [50,51] as the most relevant pathogenic mechanisms contributing to the poor obstetric outcome [52,53,54]. Each of these mechanisms is not mutually exclusive and may act together or in different combinations at different times of pregnancy. This can explain the reason why the clinical manifestations of the OAPS span from early to late miscarriages or preeclampsia [14,55]. During placentation, there is an intense trophoblast remodeling that causes the exposition of anionic phospholipids (PL) on the extracellular surface [56]. This externalization allows the interaction between PLs and β2GPI, usually expressed on the external surface of the trophoblast causing the formation of the complex anti-β2GPI/β2GPI + PL [57]. Meroni et al. suggested that a ‘second hit’ is not required in this context, since β2GPI is overexpressed on trophoblast cells and represents an easily accessible target for aPLs in the absence of the stimulation of a “second hit”. Furthermore, vascular and hormonal changes in pregnancy are suggested to resemble themselves as a ‘second hit’ [10]. Beyond trophoblasts, β2GPI has been shown to react with human stromal decidual cells and human endometrial endothelial cells, hence suggesting the heterogeneity of the effects of anti-β2GPI antibodies on placental tissue [58,59,60]. Following the binding to trophoblast cells, several consequences have been reported. In particular, aPL disrupt the annexin V shield on trophoblast cells. This protein acts as a powerful thromboregulatory molecule and seems necessary for the maintenance of placental integrity [28,52,61]. In addition, the complex anti-β2GPI/β2GPI + PL increases trophoblast cells apoptosis and decreases the trophoblast differentiation, as shown by the reduced secretion of human chorionic gonadotrophin (hCG) [51,60]. Furthermore, aPL impair trophoblast invasiveness, as observed in an in vitro Matrigel assay by inhibiting the expression/activity of the matrix metalloproteases-2 and -9, the main enzymes involved in the extracellular degradation during the trophoblast invasion [51,60,62,63]. Further effects on trophoblast functions include the inhibition of heparin binding-epidermal growth factor, HB-EGF, an important mitogen agent enhancing the trophoblast proliferation and differentiation [64].

Beyond the trophoblast cells, aPL can interact both with decidual and endometrial endothelial cells. Following the treatment of mice with aPL, the histological examination of the placentas showed both decidual necrosis, with prominent intravascular deposition of IgG and fibrin, and signs of inflammatory activation of decidual cells, as confirmed by the increased deposition of C3 and C5 [59]. In line with such observations, Berman et coll. reported that the aPL injection in pregnant mice produces an inflammatory damage in the placentas leading to fetal growth restriction and fetal resorption. Histological features in these placentas include deposition of human aPLs and mouse complement, neutrophil infiltration and local TNF secretion [65]. In a further experimental model of pregnancy loss, it has been found that the deletion of the chemokine-binding protein D6, a placental scavenger receptor controlling local inflammation by degrading the majority of inflammatory chemokines, induces fetal loss after the passive infusion with a small amount of human aPL IgG, as compared to wild-type mice or mice infused with normal IgG [66]. Altogether, these findings suggest that a local acute inflammatory response might have a role in experimental aPL-mediated fetal loss.

We previously observed that aPLs block endometrial angiogenesis both in vitro and in vivo, by inhibiting the human endometrial ECs angiogenic differentiation and the production of specific factors which are up-regulated during angiogenesis, such as vascular endothelial growth factor (VEGF) [58].

Taken together these observations suggest that the multiple aPL-mediated activities on the different cellular components of placental tissue might contribute to a final cellular insult determining a defective placentation and finally the poor obstetric outcome observed in APS women (Figure 2). Furthermore, these findings suggest that APS-associated pregnancy complications can be mediated by several distinct pathogenic events not necessarily related to the procoagulant action of aPL.

In recent years, new actors seem to be involved in the pathogenesis of pregnancy morbidity, including the release of neutrophil extracellular traps (NETosis), the extracellular vesicles (EVs) (Figure 3), and micro-RNAs (miRNAs).

NETosis is a form of programmed cell death representing an innate immune response where decondensed chromatin is spread outside the cell and acts as a trap for pathogens [67]: neutrophil extracellular traps (NETs) have antimicrobial functions, but NETosis can contribute to generate a proinflammatory state, even in the absence of an infection [68]. NETosis has been studied in several organ injuries and in reproductive disorders [69]. Neutrophil activation, including the expression of TF, the release of NETs and interleukin-8, may also be an important element of antiphospholipid-antibody-associated thrombosis [70,71,72]. In particular, NETosis seems to be activated by the interaction between aPLs and neutrophils, while in placental models, NETosis seems to begin with the interaction of aPL antibodies with placental cells. Recent evidence demonstrates that, in pregnant women with systemic lupus erythematosus, APS and preeclampsia, a higher number of NETs in the placental intervillous spaces is associated with a marked inflammation and the presence of vascular modifications [73].

Lackner et al., in a recent review, pointed out the role of extracellular vesicles [74]: APS patients have a higher number of EVs in circulation, but their role is not clear [75]. EVs are phospholipid bilayer-enclosed vesicles, very heterogeneous in size, shed from the plasma membrane, carrying antigens, cell surface receptors and ligands. According to Wu et al., aPL antibodies can induce the discharge of specific EVs from endothelial cells: these seems to be carrier of proinflammatory molecules. These EVs released in patients with APS containing IL-1β and specific single-strand-RNA molecules are able to activate the Toll-like receptor 7 (TLR7) mediated pathway. It means that these specific vesicles, released after aPL binding, amplify the inflammatory response [76]. Similar observations on the role of placenta derived extracellular vesicles were reported by Tong in 2017 [77]: the vesicles, carrying several alarmins, such as mitochondrial DNA, were able to activate endothelial cells through the TLR9 pathway.

Actually, there is increasing interest on the role of microRNA in APS. miRNAs are very short non-coding single-stranded RNA molecules. miRNAs can regulate gene expression paring with complementary messenger RNAs (mRNAs), modulating mRNA stability. miRNAs are thought to be involved in the pathogenesis of several autoimmune diseases, including APS [78]. The data were focused on two microRNAs, miR-19b and miR-20a: their downregulation in monocytes obtained from APS patients caused an increased expression of TF, contributing to generate a prothrombotic state [79,80,81].

## 6. Conclusions

Antiphospholipid syndrome still represents an important treatable cause of pregnancy morbidity. Despite the large amount of data in the literature, taken together, the above reported studies underlie the complexity and heterogeneity of the mechanisms beyond the poor obstetric outcome in APS. What is now well accepted is the ability of aPLs to directly bind the maternal and fetal side of the placental tissue and to impair placentation without necessarily activating prothrombotic phenomena. The understanding of aPL-mediated activity on the placenta is crucial for the therapeutical implications.

Standard treatment for obstetrics APS includes preconception low dose aspirin (LDA, 100 mg/day, orally) combined with low molecular weight heparin (LMWH) at prophylactic doses (0.4–0.6 mg/kg/day; 4000–6000 IU/day, subcutaneously) from the moment of the positive pregnancy test. Usually, LDA combined with LMWH treatment is continued throughout pregnancy and 6 weeks postpartum. In patients with previous thrombotic events, secondary thromboprophylaxis with full unfractionated or LMWH doses or anti-vitamin K therapy should be administered. The latter must be avoided between weeks 5 and 12 of gestation [82]. A recent metanalysis evaluated the efficacy of heparin alone or in combination with LDA for the treatment of women with RPL and aPL positivity. It was found that heparin (alone or in association with LDA) improves the live birth rate, as compared to LDA or a placebo. Such an observation was not found for LDA alone [83]. A likelihood of a good pregnancy outcome in women with APS of around 75–80% under correct management has been estimated. The initial assumption of intraplacental thrombotic phenomena leading to pregnancy morbidity led to consider heparin as a useful treatment for OAPS. However the demonstration of additional pathogenic mechanisms of aPL-mediated placental damage and the efficacy of the drug in OAPS has now suggested additional mechanisms of action. Actually, LMWH is able to block the binding of aPL to trophoblast cells and to endometrial EC. As a consequence, the LMWH is able to prevent the negative activity of aPL on trophoblast invasiveness, proliferation and differentiation, as well as on EC angiogenic processes at the endometrial level [63,84,85,86]. In addition, Girardi et al. reported that heparin has an important anti-complement activity [85].

Unfortunately, up to 20–25% of cases will not respond to standard treatment. These cases are named refractory OAPS. In this situation, other treatment schedules, mainly hydroxicloroquine (HCQ), low-prednisone dose, increasingly LMWH dose, IVIG or plasma exchange, have been considered. A possible explanation for the high proportion of refractory APS might be that not all of the mechanisms underlying aPL-mediated pregnancy complications have been clarified. Further studies and novel alternative therapies are urgently needed, among them biological therapies based on the use of immunomodulatory drugs or monoclonal antibodies which are able to prevent the aPL mediated activation of the complement and aPL pro-coagulant and pro-abortive effects.

## Figures and Tables

**Figure 1 ijms-24-03195-f001:**
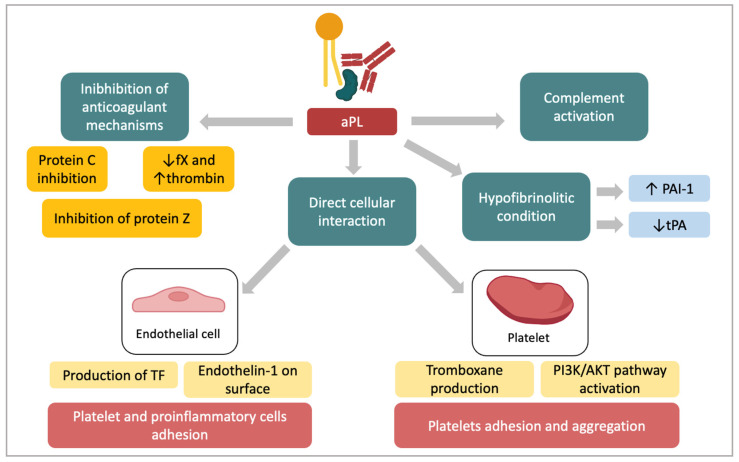
Antiphospholipid antibodies (aPL) prothrombotic activity. aPLs interact with several coagulation cascade proteins, reduce fibrinolytic activity, activate complement and have a direct effect on endothelial cells and platelets. All of these mechanisms finally contribute to the characteristic thrombotic events observed in APS. aPLs = antiphospholipid antibodies; PAI-1 = inhibitor of plasminogen type 1; tPA = activator of tissue plasminogen; TF = tissue factor; fX = factor X.

**Figure 2 ijms-24-03195-f002:**
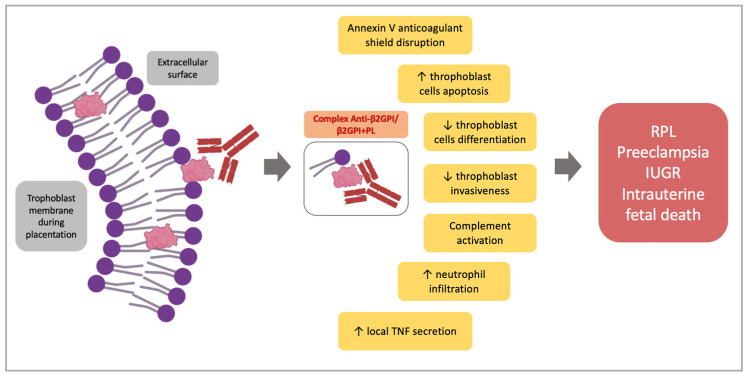
Obstetric APS mechanisms. Here are summarized the main known pathogenetic mechanisms of Anti-β2GPI and their effects on pregnancy outcomes. PL = phospholipid; TNF = tumor necrosis factor; Anti-β2GPI = β2 glicorpotein1-antibodies; β2GPI = β2 glycoprotein 1; RPL = recurrent pregnancy loss; IUGR = intrauterine growth restriction.

**Figure 3 ijms-24-03195-f003:**
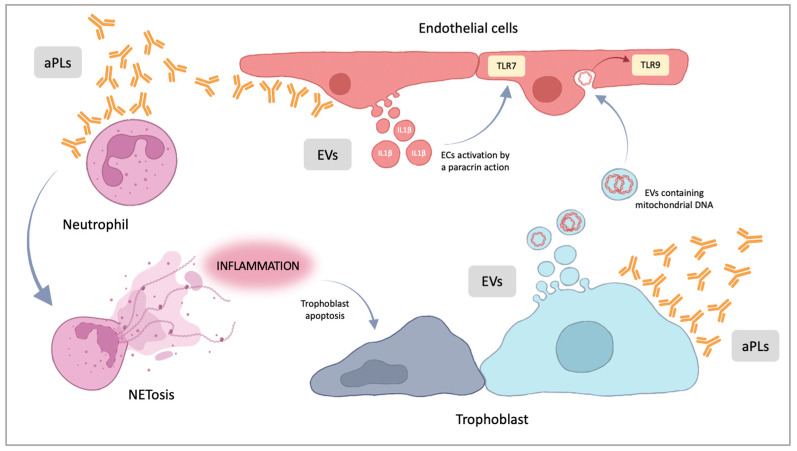
NETosis and extracellular vesicles in obstetrical APS pathogenesis. aPLs can lead to neutrophil NETosis promoting an inflammatory state in placental stroma that can induce trophoblast apoptosis. aPLs can promote endothelial cells (ECs) EVs generation, containing IL-1β and ss-RNA that can activate, by a paracrine action, other ECs via TLR7. EV shedding is promoted also by aPLs interaction with trophoblast. Trophoblast EVs can contain several alarmins, such as mitochondrial DNA, that can activate endothelial cells via TLR9. aPLs = antiphospholipid antibodies. EVs = extracellular vesicles; TLR7 = Toll-like receptor 7; TLR9 = Toll-like receptor 9.

**Table 1 ijms-24-03195-t001:** Sydney criteria for APS.

1. Vascular thrombosis	≥1 clinical episode of thrombosis (arterial or venous), in any tissue or organ.
2. Pregnancy morbidity	≥1 morphologically normal fetal loss, ≥10th week of gestation, or ≥1 premature birth of a normal neonate before the 34th week because of: (i) eclampsia or severe preeclampsia or (ii) due to placental insufficiency, ≥3 unexplained consecutive spontaneous abortions < 10th week of gestation (maternal or paternal anatomic, hormonal or chromosomal causes excluded).
3. Laboratory criteria	Lupus anticoagulant (LA) present in plasma, in ≥2 samples, Anticardiolipin (aCL) antibody IgG and/or IgM present in medium or high titer, on ≥2 occasions at a distance of 12 weeks, Anti-β2 glycoprotein-I (anti-β 2GPI) antibody IgG and/or IgM present in medium or high titer, on ≥2 occasions at a distance of 12 weeks.
Subclassification of APS patients (according to aPL positivity)	Type I: >1 laboratory criterion present (any combination), Type IIa: LA antibodies only, Type IIb: aCL antibodies only, Type IIc: anti-β2 glycoprotein-I antibody only. Multiple aPL positivity is correlated with a more serious course of the syndrome.

## Data Availability

Not applicable.

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
