# Peer review of "Antiphospholipid Syndrome in Pregnancy: New and Old Pathogenetic Mechanisms"

_ijms, 2023, doi:10.3390/ijms24043195_

Round 1
Reviewer 1 Report
Very interesting and important work that documents the implication of APL in obstetric complication without thrombosis as a main mechanism.
I have an unique suggestion to advance some directions toward the treatment possibilities that those important work opens.
Reviewer 2 Report
1) Abstract. L19-21. The most frequent complications in OAPS are recurrent pregnancy losses end premature birth due to placen- tal insufficiency or severe Preeclampsia. In the recent years Vascular APS (VAPS) and Obstetric APS (OAPS) have been described as two different clinical entities. Please, correct the typo and explain the acronym Obstetric APS (OAPS) the first time you use it.
2) Abstract. L26-27. Furthermore, new actors seem to play a role in pathogenesis of OAPS including Extracellular Vesicles, micro-RNAs and the release of neutrophil extracellular traps. Please, underline the aim of the study
3) Introduction. L33-36. Concisely, APS is characterized by persistent positivity for antiphospholipid antibodies [aPL, includ- ing Lupus Anticoagulant (LA), anticardiolipin (aCL) and anti-β2-glycoprotein1 antibod- ies (anti-β 2GPI)], thrombotic events and/or severe pregnancy morbidity [1,2]. In order to improve and discuss the previously described points, important references are needed to be added, such as:
a- Differences in Antiphospholipid Antibody Profile between Patients with Obstetric and Thrombotic Antiphospholipid Syndrome. Int. J. Mol. Sci. 2022, 23, 12819. https://doi.org/10.3390/ijms232112819
b-Asymmetric Dimethylarginine Is a Marker of Endothelial Dysfunction in Thrombotic Antiphospholipid Syndrome Patients. Int. J. Mol. Sci. 2022, 23, 12309. https://doi.org/10.3390/ijms232012309
4) APS can be defined as “primary APS” when it is isolated, or “secondary APS” when oc- 43 curs associated to other autoimmune diseases, mainly Systemic Lupus Erythematous 44 (SLE) [5]. Cervera et al found that the 53.1% of patients had “primary APS”, 36.2% had 45 Systemic lupus erythematosus, 5.0% had Lupus-like syndrome, 2.2% had Primary 46 Sjögren’s syndrome, 1.8% had Rheumatoid arthritis, 0,7% had Systemic sclerosis, 0,7% 47 had Systemic vasculitis and 0,5% had Dermatomyositis [6]. Please, improve this paragraph and add several references:
a- Correlation between Potential Risk Factors and Pulmonary Embolism in Sarcoidosis Patients Timely Treated. J Clin Med. 2021;10(11):2462. Published 2021 Jun 2. doi:10.3390/jcm10112462
b- Kikuchi-Fujimoto's disease associated with systemic lupus erythematous: difficult case report and literature review. Lupus. 2014;23(9):939-944. doi:10.1177/0961203314530794
c- IgA Antiphospholipid Antibodies in Antiphospholipid Syndrome and Systemic Lupus Erythematosus. Int. J. Mol. Sci. 2022, 23, 9432. https://doi.org/10.3390/ijms23169432
5) Introdusction L57-59. In the present review we will focus on OAPS, and in particular on the pathogenesis of pregnancy complications observed in the syndrome. Please improve the description of study aim and underline the novelti of this paper.
6) Figure 1. L159-165. Antiphospholipid antibodies (aPL) prothrombotic activity. aPLs interact with sev- eral coagulation cascade proteins, reduce fibrinolytic activity, activate complement and have a di- rect effect on endothelial cells and platelets. All these mechanisms finally contribute to the charac- teristic thrombotic events observed in APS. aPLs = antiphospholipid antibodies; PAI-1= Inhibitor of Plasminogen type 1; tPA= activator of tissue Plasminogen; ECs = endothelial cells; TF = tissue factor; fX = factor X. The figure are clear and interesting.
7) 6. Conclusions L278-285. Antiphospholipid Syndrome still represents an important treatable cause of pregnancy morbidity. Despite the large data in the literature, taken together, the above reported stud- ies underlie the complexity and heterogeneity of the mechanisms beyond the poor obstet- ric outcome in APS. What is now well accepted is the ability of aPLs to directly bind ma- ternal and fetal side of the placental tissue and to impair placentation without necessarily activating prothrombotic phenomena. The understanding of aPL-mediated activity on placenta is crucial for the therapeutical implications. Please, improve the conclusione and underline the clinical implications of this manuscript
